# Radiologist workforce challenges and the burden of image interpretation in Ghana: Perspectives of frontline doctors and implications for healthcare delivery

Bismark Ofori-Manteaw[1¤], Eric Kwasi Ofori[2], Klenam Dzefi-Tettey[3], Jacob Leonard Ago[4], Seth Kwadjo Angmorterh[1], Frempong Acheampong[5]*

1 Department of Medical Imaging, University of Health and Allied Sciences, Ho, Ghana, 2 Department of Imaging Technology and Sonography, University of Cape Coast, Cape Coast, Ghana, 3 Department of Radiology, School of Medicine, University of Health and Allied Sciences, Ho, Ghana, 4 School of Health and Biomedical Sciences, RMIT University, Bundoora, Australia, 5 School of Basic and Biomedical Sciences, University of Health and Allied Sciences, Ho, Ghana

¤ Current address: Medical Radiation Science Discipline, Faculty of Science and Health, Charles Sturt University, Wagga Wagga, Australia
* facheampong@uhas.edu.gh

## Abstract

Interpreting radiological images, a primary responsibility of radiologists, is crucial for accurate diagnosis and informed clinical decisions. However, many low-and-middle-income countries (LMICs) face severe radiologist shortages, leading to diagnostic delays, uncertainty, and potential compromises in patient care. This study explored the perspectives of frontline medical doctors on the challenges posed by radiologist shortages in Ghana concerning image interpretation and its impact on healthcare delivery and patient care. A qualitative descriptive study was conducted using semi-structured interviews with eight medical doctors working across public and private healthcare facilities at different levels of the Ghanaian health system. Data were thematically analysed with NVivo 14, following Braun and Clarke's reflexive thematic analysis framework. Three key themes emerged: the vital role of radiological imaging in clinical practice, difficulties in accessing radiological reports, and suggested strategies to address the challenges posed by radiologist shortages. Subthemes identified include barriers such as delayed access to radiologist reports, underlying causes for report unavailability, insufficient training in image interpretation, and clinician burnout. Participants proposed expanding radiology training, strengthening radiology education in undergraduate medical curricula, involving trained radiographers in preliminary image evaluation, and integrating teleradiology and artificial intelligence technologies as potential solutions to these challenges. Radiologist shortages in Ghana significantly affect the availability and quality of image interpretation services, which impacts diagnostic accuracy and patient care. Addressing this gap requires a multifaceted approach involving radiologist workforce development and enhancing

**Data availability statement:** Due to ethical and confidentiality restrictions related to the qualitative interview data, the full interview transcripts cannot be made publicly available. De-identified excerpts relevant to the study findings are included within the paper. Additional de-identified data may be made available upon reasonable request and subject to approval by the University of Health and Allied Sciences Research Ethics Committee (UHAS-REC) via https://ihr.uhas.edu.gh/research-center/uhas-research-ethics-committee-uhas-rec. Data are stored in accordance with the University's Research Data Management Policy to ensure confidentiality and security.

**Funding:** The authors received no specific funding for this work.

**Competing interests:** The authors have declared that no competing interests exist.

the image interpretation skills of front-line medical practitioners. Efforts should include enhancing radiology training in undergraduate medical education and expanding access to remote reporting technologies particularly in underserved areas.

## Introduction

Radiological imaging is crucial in modern clinical practice for diagnosis, treatment planning, monitoring disease progression, and guiding interventions across various medical specialities [1–4]. Radiologists play a crucial role in enhancing diagnostic accuracy through their expertise in interpreting medical images, which significantly contributes to improved patient outcomes. The comprehensive reports they produce enable clinicians to identify subtle imaging findings, reducing the likelihood of misdiagnosis and facilitating informed decisions about the most suitable treatment options [1,5]. The value of timely radiological reporting has been highlighted in several studies [2,6–9] linking delayed or absent reports to inappropriate clinical management and adverse events. In emergency care settings, even brief reporting delays can compromise treatment decisions, with potentially life-threatening consequences [2].

Healthcare systems globally are faced with acute radiologist shortages [1,8,10–14] amid growing demand for imaging services, resulting in reporting delays and backlogs of X-ray images [1,12,15,16]. In Australia, for instance, emergency department X-ray reports could be delayed by more than four hours [12]. In South Africa, X-ray reporting could take up to 10 days, with most radiographs unreported [16]. The golden rule, however, is that all radiological investigations should be accompanied by radiological reports to enhance the accuracy of patient diagnoses [17].

In low- and middle-income countries (LMICs), significant gaps in diagnostic workforce hinder the achievement of universal health coverage and exacerbate systemic health inequities [4,10,18] Many under-resourced countries have radiologist densities far below international benchmarks [8,9,19]. Ghana has a poor radiologist-to-population ratio of 0.31 per 100,000 [8]. To contextualise the scale of the problem, findings from a recently published national facility audit conducted as part of a broader doctoral study provide important background. That study reported that 53.8% of X-ray facilities in Ghana had no radiologist on site, with only 19.7% having full-time radiologist coverage and 16.2% relying on remote reporting. Among facilities with access to radiologists, emergency cases were commonly reported within 24 hours, whereas non-emergency examinations frequently experienced delays of two to three days. Notably, over half of facilities indicated that all radiographs produced remained unreported, despite many performing more than 10,000 general radiographic examinations annually [20]. These findings highlight the magnitude of reporting gaps within the Ghanaian health system and underscore the clinical context within which frontline doctors are required to make diagnostic decisions. While these quantitative data describe the scale of the challenge, they do not capture how frontline doctors experience, navigate, and respond to these diagnostic gaps in everyday practice.

Consequently, X-ray images are often interpreted only upon specific request, compromising timely diagnosis and patient care. In the absence of radiologists, interpretation typically falls to non-specialist medical officers, house officers, or radiographers, whose training and confidence in diagnostic interpretation vary widely [9,18,21]. In the Ghanaian context, radiographers are not authorised to provide formal diagnostic reports, and their involvement in image interpretation is generally limited to informal support or preliminary image evaluation (PIE) rather than clinical reporting. The contrast with high-income countries is stark: the EU averages 12 radiologists per 100,000 population, Canada 7, and the UK [5,8,15]. These disparities highlights global inequities in radiology workforce distribution, posing serious risks to timely care in countries with fragile health systems. Addressing radiologist shortages is not only a clinical necessity but also a global development priority aligned with Sustainable Development Goals 3 and [10,22,23].

Radiologist shortages have far-reaching implications. Doctors are often forced to interpret complex images with limited training, increasing the risk of diagnostic errors [4,9,11,24]. These errors may lead to inappropriate treatment, delayed care, and potential harm or death [9,25–27]. The burden on junior doctors to make critical decisions without radiologist input can also contribute to stress and burnout [28] Moreover, the lack of timely radiological support delays patient flow in emergency and outpatient settings and undermines system efficiency [9,29]. Rural and underserved facilities without radiologists face major challenges, resulting in poorer diagnostic services compared to urban centres [8]. This urban-rural gap deepens structural inequalities in access to quality care, especially in countries with known regional disparities in radiologist health workforce distribution [8]. Although radiologist workforce challenges are well documented globally [1,4,8,10–13], empirical studies that explores the experiences of frontline doctors is limited [16]. Little is known about how frontline doctors cope with these shortages, their views on potential solutions, and their training needs in image interpretation. Understanding these perspectives is crucial for informing workforce planning, education, and sustainable service delivery.

Also, in many medical training programmes, radiology education for medical students is relatively limited and often delivered as short modules within broader clinical curricula [27,30,31]. Training typically focuses on basic principles of imaging and introductory interpretation of common radiographs, particularly chest and musculoskeletal X-rays. However, opportunities for structured practical training in image interpretation may be constrained in settings with limited radiology workforce capacity [8,20]. In resource-constrained health systems such as Ghana, radiologist shortages may further restrict the time available for clinical teaching and supervision, as available radiologists are often required to prioritise service provision, including reporting workloads [32]. Consequently, newly qualified doctors may enter practice with variable confidence and competence in image interpretation, particularly when radiologist support is unavailable [31]. This study explored the experiences and perspectives of frontline medical doctors in Ghana regarding radiologist shortages, with a particular focus on their implications for image interpretation, healthcare delivery, and patient care.

## Materials and methods

### Ethical clearance

The study received ethical approval from the University of Health and Allied Sciences Research Ethics Committee (UHAS-REC A.6 [3] 22–23). All participants provided verbal consent before participating in the study.

### Study design and participants

This study employed a qualitative descriptive design to explore frontline doctors' experiences and perspectives on radiologist shortages in Ghana and their impact on patient care following. This qualitative study forms part of a broader multi-phase doctoral study; quantitative findings relating to radiologist availability, reporting turnaround times, and volumes of unreported radiographs have been published separately and are referenced here for contextual framing only. A qualitative

approach enabled in-depth understanding of frontline doctors' daily practices, decision-making, and proposed solutions in settings with limited radiologist support [33].

Participants (medical doctors) were recruited using purposive sampling, aimed at achieving maximum variation in terms of healthcare facility level and geographic location across Ghana. Eligible participants included frontline medical doctors who routinely request radiological examinations as part of patient management. Potential participants were identified through professional networks and selected based on their clinical roles and experience in requesting and interpreting radiological examinations. These included senior medical officers (SMOs), medical officers (MOs), senior house officers (SHOs), and house officers (HOs) working across district, regional, and tertiary healthcare facilities.

Ghana's healthcare system is structured into four tiers. At the base is the primary healthcare level, comprising basic clinics and community-based services that serve as the first point of call for patients. The second level (district hospitals) functions as the first referral facilities, while the regional level offers specialized medical services and supports the lower tiers. At the top is the tertiary level, which provides advanced, complex medical care and serves as the apex referral system within the country's healthcare structure [34]. This variation enabled a broader exploration of the issues across diverse resource settings, increasing the transferability of findings. Specialists and consultants were excluded to focus on frontline doctors who often make independent diagnostic decisions without formal radiology reports. Potential participants were identified through professional networks and contacted through telephone calls or email invitations. Follow-up contact was made to confirm willingness to participate, and lack of response after follow-up was interpreted as non-participation. Only fourteen potential participants initially agreed to participate in the study.

### Data collection

Data were collected semi-structured interviews conducted in English between 1st September 2023 and 20th October 2023. The interviews were conducted by the lead researcher, an academic who has experience in image interpretation education and training in conducting semi-structured interviews. Participants were briefed on the researcher's professional background and the purpose of the study prior to the interviews, which helped establish trust and facilitate open dialogue during the data collection process. Interviews were conducted on a one-to-one basis, allowing participants sufficient opportunity to expand on their experiences and viewpoints and to offer detailed, context-rich accounts. Interviews explored participants' experiences with accessing radiology reports, the clinical impact of report unavailability on patient care, and suggestions for improving radiological support. Verbal consent was obtained from all participants and audio recorded, with participants informed that they could withdraw from the study at any time. Each interview was conducted via telephone, audio-recorded and lasted approximately 20–35 minutes. A literature-informed interview guide was developed based on the study objectives and previous research on radiology workforce shortages, image interpretation practices, and access to radiological reporting in resource-limited settings [9,11,21,35]. The guide included questions exploring the use of radiology in clinical practice, challenges in accessing radiological reports, radiology training and doctors' experience in image interpretation, report turnaround times, perspectives on radiographer role extension, and potential strategies to improve access to radiological support. The credibility of the interviews was strengthened through prolonged engagement, iterative questioning, member checking, and peer debriefing with an experienced qualitative researcher. This approach ensured consistency in the data collection process and minimised interviewer bias.

In qualitative research, sample size is guided by the concept of information power [36], with adequacy assessed by the richness of data in relation to the study's aim, sample specificity, and quality of dialogue. In this study, interviews were analysed concurrently with data collection to assess the emergence of new concepts. The final sample size was considered sufficient at the point of data saturation, where no new insights or themes were identified and responses became repetitive across interviews [37,38], indicating that saturation had been reached. This indicated that the data collected

were both comprehensive and adequate to address the study objectives. In all, eight medical doctors were interviewed at the point where thematic saturation was reached.

## Data analysis

Identifiable information was anonymised during transcription, analysis, and reporting. Interviews were transcribed verbatim and analysed thematically using NVivo 14 software to support data organisation and coding, following Braun and Clarke's six-phase framework [39]. The analysis began with familiarisation with the data through repeated readings of the interview transcripts to gain an in-depth understanding of participants' accounts. Initial coding was undertaken independently by two members of the research team to enhance analytical rigour using a combination of inductive and deductive coding strategies. Codes were subsequently collated into potential themes representing patterned responses relevant to the research objectives. These themes were reviewed iteratively to ensure coherence within themes and alignment with the coded extracts and the full dataset. Through a reflexive process of refinement, themes were clearly defined and named to capture the essence of the participants' perspectives. Peer debriefing and regular team discussions were also used to enhance analytical rigour and credibility throughout the process. Discrepancies in coding or theme development were resolved through discussion and consensus. The study was conducted and reported in accordance with the Consolidated Criteria for Reporting Qualitative Research (COREQ) checklist [40,41].

## Results

Eight participants were interviewed: three (3) senior medical officers (SMOs), two (2) medical officers (MOs), two (2) senior house officers (SHOs), and one (1) house officer (HO). Four participants were based in district hospitals, two in regional hospitals, and two in tertiary-level hospitals. The mean age of participants was 33±4 years, and their clinical experience ranged from 1 to 10 years.

Analysis of the interview data generated three key themes and nine subthemes, as illustrated in Fig 1.

### Theme one: Relevance of radiological imaging in clinical practice

Theme one highlights the central role that radiological imaging plays in routine clinical decision-making. Participants consistently described imaging as an essential diagnostic tool used across multiple clinical contexts.

*Sub-theme 1: Frequency of imaging requests.* Participants indicated that radiological imaging forms a routine component of clinical decision-making across different levels of care. Several doctors described requesting imaging on a daily basis as part of patient assessment and management.

*"Quite a lot; I can't count, almost every day I request Xray imaging." (DR05)*

According to them, requests for imaging were more frequent at the accident and emergency and outpatient departments.

*"Very often [but] it depends [on] where you are. …but at the accident and emergency and the OPD, you tend to request a lot"* (DR06).

**Sub-theme 2: The need for formal radiologist reports.** While some doctors expressed confidence in interpreting basic radiographs, most underscored the added value of formal reports, particularly in complex or acute cases.

*"In as much as I may find some one or two anomalies on the X-ray, the radiologist interpretation goes a long way to help as well"* (DR04).

*"It is very important because it increases quality care and decision-making, especially in acute cases…"* (DR07).

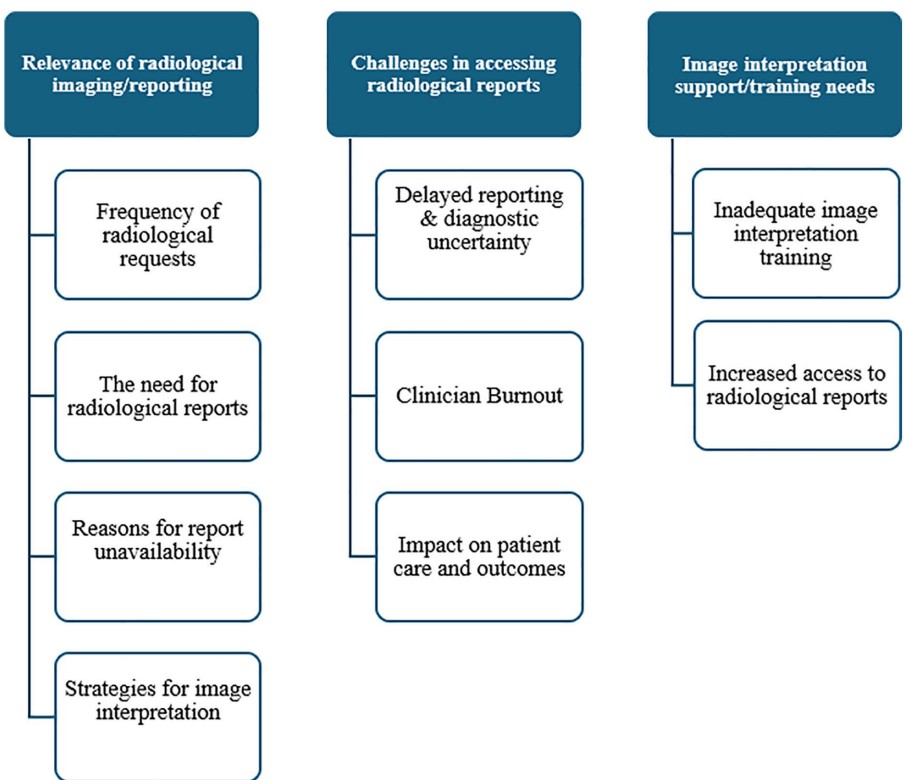

**Fig 1. Key themes and subthemes Identified.**

Inadequate undergraduate radiology training was also noted, with most doctors advocating for radiologist support to ensure diagnostic reliability.

*"… But even when you enter the field, because you haven't studied radiology as a full program, you don't really have all the answers. You might think you see something when it's something else" (DR01).*

**Sub-theme 3: Reasons for report unavailability.** Despite recognizing the value of formal radiologist reports, participants noted that radiographs were often delivered without accompanying interpretations. This was attributed to assumptions that senior medical officers could interpret images independently, as well as uncertainty surrounding reporting protocols.

*"The specialists are trained to read the images. So, they interpret it on their own. They only request for [a report] when they are having difficulties." (DR02)*

*"I don't know if I have to request [for] a report or if they are supposed to do that, but honestly, it doesn't come with any radiological report." (DR04)*

Another reason was the limited access to radiologists. Participants explained:

*"One of the causes would be the number of radiologists we have; I only know of a few in the hospital." (DR01)*

**Sub-theme 4: Strategies used in the absence of radiological reports.** When radiologist reports aren't available, frontline doctors adopt various strategies.

*"I use my knowledge from school and what I learnt during my radiology rotation. Then sometimes [too], I discuss with my senior colleagues." (DR03).*

*"If I see something that I'm not too sure about, I ask a senior colleague…because getting the radiologist is a bit difficult." (DR01)*

**Theme two: Challenges in accessing radiological reports**

The participants emphasized several challenges they face in accessing X-ray interpretation.

*Sub-theme 1: Delayed reporting.* Participants reported noticeable variation in reporting turnaround times between facilities, with some doctors indicating that reports were sometimes delayed or unavailable at the time of clinical decision-making. One participant explained:

*"Some come with reports and some too don't or they report later. At least some take a day." (DR07)*

Another elaborated on these disparities by comparing experiences across different locations:

*"In [XYZ], I used to get report within 72 hours, but in [ABC], it takes 24 to 48 hours to get radiological report." (DR03)*

*Sub-theme 2: Clinician burnout.* The burden of interpreting radiographs without adequate support contributed to doctors' stress and burnout. One participant reflected on the emotional toll of this situation:

*"It's frustrating when you don't understand what you're looking at and can't get the help you need." (DR05)*

Another participant highlighted the added stress of seeking radiologist interpretation while simultaneously managing competing clinical demands:

*"Looking for specialists to help interpret the radiographs can be stressful, particularly when other patients are waiting to be attended to." (DR04)*

**Sub-theme 3: Impact on patient care and outcomes.** Frontline doctors noted that in the absence of radiologist reports, diagnostic uncertainty increased.

*"…Once you don't know, you resort to guesswork, which can lead to misdiagnosis." (DR05)*

Some participants also linked the lack of timely radiological report to delayed treatment decisions and poorer clinical outcomes.

*"It affects the quality of care, because if there's probably something in there which you could have identified, an immediate action would have been taken." (DR07).*

These challenges were considered most acute in under-served regions. One participant stated:

*"… They [junior doctors] cannot make any major decisions because they'll be waiting for a report from a qualified person, and so it'll delay their interventions." (DR03)*

**Theme three: Support and training needs**

The third theme highlights the critical need for enhanced support and training in X-ray interpretation among frontline doctors.

*Sub-theme 1: Inadequate image interpretation training.* Several participants highlighted inadequate radiology training in medical school curricula, which limited doctors' ability to independently interpret imaging findings.

*"We were taught how to interpret these images, mostly chest X-rays but it was a semester course." (DR01)*

*"With the curriculum I believe it's inadequate because the duration is very short." (DR08)*

**Sub-theme 2: Image interpretation strategies/support.** Participants proposed strategies to address radiologist shortages, including improved radiology education during medical training and regular continuing professional development (CPD) to enhance image interpretation skills among non-specialists.

*"The basics at the MBCHB level should be reinforced so that it encourages more people to enter radiology training after school." (DR02)*

*"In medical school you just do it [radiology] for a short period… So, doctors need refresher courses, so that at least they can handle the basics and refer complex cases to radiologists." (DR06)*

Some participants suggested leveraging the expertise of trained radiographers to provide preliminary image comments, particularly in facilities without radiologists to support junior doctors.

*"Some of them [radiographers] assist us interpret images, if they are given proper support, they can really assist us." (DR07)*

Technology such as Artificial intelligence (AI), teleradiology and remote reporting systems were also endorsed as viable options.

*"Facilities could rely on artificial intelligence and teleradiology to overcome some of the challenges in accessing radiological reports." DR01*

As a long-term measure, some participants advocated for increased radiologist training and decentralization particularly in regional and district-level hospitals.

*"…There should be more training of specialists [radiologists] and then after, we should decentralize them to all the various regions." (DR03)*

On the whole, participants emphasised the need for additional training opportunities and institutional support to improve clinicians' ability to interpret radiological images when specialist input is not readily available.

## Discussion

This study aimed to explore the experiences and perspectives of frontline medical doctors in Ghana regarding radiologist shortages and their implications for image interpretation, healthcare delivery, and patient care. The findings highlight substantial challenges in accessing timely radiological reports, which compromise diagnostic decision-making and place additional responsibility on frontline doctors.

## Role of radiological reporting in clinical care

Participants emphasized the critical role of radiological reports in clinical practice, consistent with global literature identifying radiologist interpretation as essential to high-quality care [2,6–9]. While doctors regularly relied on imaging to inform decisions, many acknowledged limited training in image interpretation and expressed a strong preference for formal radiologist input, particularly in complex cases. In low-resource settings like Ghana, where diagnostic resources are limited, radiological reports play a pivotal role in clinical decision-making. However, this study found that X-rays are often provided without accompanying reports, a trend also observed in other low- and middle-income countries (LMICs) facing radiologist shortages [9,11,13,42,43]. This has led to task-shifting and increased reliance on clinician self-interpretation, which is associated with a higher risk of misdiagnosis, especially among junior doctors with limited experience [1,5]. Ensuring the consistent availability of radiologist reports is therefore essential for improving diagnostic accuracy, supporting clinical learning, reducing uncertainty, and enhancing patient outcomes.

## Barriers to accessing radiology reports

Best practices recommend that radiological reports be delivered within 4 hours for urgent cases and 24 hours for non-urgent ones to support timely clinical management [44]. However, this goal is often unattainable in settings with a limited radiologist workforce [12]. Prolonged turnaround times for X-ray interpretation remain a global challenge to effective healthcare delivery [45,46]. This study identified several structural and systemic barriers to timely report access, key among them being the acute shortage of radiologists, particularly in district and regional hospitals.

This gap results in delayed or absent reporting, compelling doctors to rely on limited imaging expertise, a trend consistent with evidence from sub-Saharan Africa, where radiologist-to-population ratios are critically low [4,8,9,19,47]. Even in facilities with radiologists, heavy workloads and competing duties often delay reporting [1]. Participants expressed frustration over diagnostic uncertainty and indicated instances where delays adversely affected patient outcomes. These findings imply that workforce shortages are not only numerical and but also functional [48]. Moreover, the lack of support systems, standardized reporting protocols, and the uneven geographic distribution of radiologists further compromise equitable access to quality care [49]. It is important to note that the interpretation challenges described in this study largely relate to plain radiography, which constitutes the most commonly available imaging modality in many Ghanaian healthcare facilities.

## Coping strategies and suggested solutions

Frontline doctors reported relying on strategies such as consulting senior colleagues and drawing on limited undergraduate training in the absence of formal X-ray reports. However, these were seen as temporary solutions inadequate for ensuring diagnostic accuracy, especially in complex cases. The findings highlight the urgent need to expand CPD in image interpretation, consistent with previous studies [9,50]. Participants also stressed the importance of strengthening radiology education at the undergraduate level. Although included in medical curricula, radiology exposure is often minimal, leaving many graduates underprepared for independent interpretation [27]. Enhancing radiology instruction through structured clinical rotations and supervised practice could help develop foundational skills and reduce reliance on overburdened radiologist services.

The findings of this study also highlight concerns regarding the level of radiology training and image interpretation preparedness among referring doctors. This issue is not unique to Ghana. Studies from several countries have reported that radiology education is often underrepresented within undergraduate medical curricula, resulting in many medical graduates feeling insufficiently prepared to interpret basic radiographic images independently [31]. Similar concerns have been raised in broader reviews of radiology education, which indicate considerable variability in how imaging is taught across medical schools and limited structured training in image interpretation for non-radiologist doctors [51]. These findings

align with the experiences reported by participants in the present study, suggesting that strengthening radiology education during undergraduate medical training may be an important component of addressing diagnostic challenges in settings affected by radiologist shortages.

Technology was widely proposed as a potential solution to radiologist shortages [10,52]. Teleradiology was especially highlighted for their ability to mitigate geographic disparities in radiologist availability, an observation consistent with literature demonstrating their feasibility and impact in LMICs [2,10,53,54]. Establishing reliable remote reporting systems could improve access to specialist interpretation and help address uneven workforce distribution [55]. However, successful teleradiology implementation requires coordinated national strategies, sustained investment in digital infrastructure, and capacity-building efforts to ensure effective and ethical use [56–58]. Participants also saw AI as a promising tool, particularly in low-resource settings, but emphasized the need for careful clinical, ethical, and contextual integration to ensure it supports rather than replaces professional judgment [2,56].

Participants advocated for radiographer involvement in image interpretation as a potential strategy to support doctors in facilities without radiologists. Emerging developments across Africa suggest a growing interest in radiographer role extension in image interpretation and reporting. For example, Uganda and Zimbabwe have implemented a structured training programme in plain film image interpretation as part of efforts to strengthen healthcare delivery [59]. In addition, there are ongoing discussions and exploratory initiatives in other countries within the region [11,13,16,53,60]. However, the extent to which these developments represent fully established clinical reporting roles remains unclear due to limited published evidence. Nevertheless, these regional trends highlight a broader recognition of the potential role of radiographers in supporting diagnostic pathways, particularly in contexts affected by radiologist workforce shortages.

While formal radiographer clinical reporting remains limited in many African countries, several studies have also explored the potential role of radiographers in preliminary image evaluation as a strategy to support doctors in settings affected by radiologist unavailability. In these contexts, PIE typically involves radiographers providing initial written comments or opinions on radiographs to assist clinical decision-making [14,18,61]. Among the study participants, this model was viewed as a practical means to bridge gaps in image interpretation in remote and underserved areas [13,18,32,61]. Participants however emphasized that such approach would require clear regulatory frameworks, defined role boundaries, and targeted training to ensure competence and patient safety [35].`

Furthermore, expanding the radiology workforce is a critical long-term strategy to address systemic shortages. Increasing the number of trained radiologists, alongside targeted incentives for deployment to underserved regions, could help reduce geographic disparities in access. Strengthening district and regional hospitals with modern radiological equipment may further attract and retain radiologists. Additionally, investing in CPD and clear career advancement pathways can enhance retention and support workforce sustainability. These measures collectively have the potential to strengthen diagnostic capacity, reduce delays in care, and improve patient outcomes across Ghana's health system.

### Implications for health workforce planning and system reform

Addressing radiologist workforce shortages requires more than increasing personnel, it calls for a comprehensive, systems-level approach to workforce planning, training, and task allocation. The challenges highlighted in this study point to broader systemic flaws within the healthcare system. These gaps hinder timely access to imaging interpretation services and place excessive pressure on frontline doctors, contributing to burnout and impaired clinical decision-making. Building a resilient diagnostic system necessitates strengthening radiology education, leveraging technologies such as teleradiology and AI, and enhancing the image interpretation skills of front-line medical practitioners. Sustainable solutions must be embedded within national health workforce frameworks and broader service delivery reforms to ensure equitable access to radiology services and enhance overall health system performance in resource-limited settings.

## Study limitations

This study has several limitations that should be acknowledged. The qualitative design and relatively small sample size may limit the generalisability of the findings across all clinical settings in Ghana. Although data saturation was achieved, the purposive sampling approach may introduce selection bias, as participants with particularly strong views or experiences may have been more likely to respond. The exclusion of radiologists' perspectives is another important limitation, as their insights could have enriched the analysis and offered a more comprehensive understanding of the systemic barriers and potential solutions to radiological reporting challenges. In addition, the study relied solely on self-reported data from doctors without triangulation through observation or institutional records, which may affect the objectivity of some findings. Future research should incorporate a broader range of stakeholder views to enhance evidence-based policy and practice. Additionally, future studies may also consider examining subgroup-specific experiences across different levels of healthcare or clinical roles to further enhance contextual understanding.

## Conclusion

Radiologist shortages in Ghana have significant implications for timely image interpretation, diagnostic accuracy, and overall healthcare delivery. The findings highlight the challenges frontline medical doctors face in accessing radiological reports and the resulting impact on clinical decision-making and patient care. Addressing these challenges requires a multifaceted approach that includes strengthening the radiologist workforce, improving radiology training for referring clinicians, and expanding access to technologies such as teleradiology. Importantly, the findings also highlight the potential value of radiographer role extension, including preliminary image evaluation and the future development of reporting radiographers, as part of broader strategies to improve diagnostic imaging services in resource-constrained settings.

## Author contributions

**Conceptualization:** Bismark Ofori-Manteaw, Eric Kwasi Ofori, Klenam Dzefi-Tettey, Seth Kwadjo Angmorterh, Frempong Acheampong.

**Data curation:** Bismark Ofori-Manteaw, Eric Kwasi Ofori, Jacob Leonard Ago, Frempong Acheampong.

**Formal analysis:** Bismark Ofori-Manteaw, Klenam Dzefi-Tettey, Jacob Leonard Ago.

**Investigation:** Bismark Ofori-Manteaw, Jacob Leonard Ago, Frempong Acheampong.

**Methodology:** Bismark Ofori-Manteaw, Eric Kwasi Ofori, Klenam Dzefi-Tettey, Jacob Leonard Ago, Seth Kwadjo Angmorterh, Frempong Acheampong.

**Project administration:** Bismark Ofori-Manteaw, Eric Kwasi Ofori, Klenam Dzefi-Tettey.

**Supervision:** Eric Kwasi Ofori, Klenam Dzefi-Tettey, Seth Kwadjo Angmorterh.

**Validation:** Eric Kwasi Ofori, Seth Kwadjo Angmorterh.

**Writing – original draft:** Bismark Ofori-Manteaw.

**Writing – review & editing:** Eric Kwasi Ofori, Klenam Dzefi-Tettey, Jacob Leonard Ago, Seth Kwadjo Angmorterh, Frempong Acheampong.

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
