## [Decision Letter · Decision Letter 0]

26 Nov 2025

PGPH-D-25-02975

Radiologist workforce challenges and the burden of image interpretation in Ghana: Implications for healthcare delivery and patient care

Dear Dr. Acheampong,

Thank you for submitting your manuscript to PLOS Global Public Health. After careful consideration, we feel that it has merit but does not fully meet PLOS Global Public Health’s publication criteria as it currently stands. Therefore, we invite you to submit a revised version of the manuscript that addresses the points raised during the review process.

Please note that we have only been able to secure a single reviewer to assess your manuscript. We are issuing a decision on your manuscript at this point to prevent further delays in the evaluation of your manuscript. Please be aware that the editor who handles your revised manuscript might find it necessary to invite additional reviewers to assess this work once the revised manuscript is submitted. However, we will aim to proceed on the basis of this single review if possible.

The comments from the reviewer can be found below. In addition to their comments we would like to request the following:

• Please include more detail on how participants were selected/recruited in the methods section.

• Please upload a copy of the interview guide & COREQ checklist as supplementary files.

• Please provide more detail on how the themes were developed.

• Include further discussion of the study limitations.

• If available, authors should provide the data requested by reviewer 1, or else at a minimum present more local/national/regional statistics in the Introduction/Discussion on the topics cited by the reviewer.

Please review these comments and those provided in Reviewer 1's assessment below and make the appropriate revisions to your manuscript alongside a point-by-point response to the concerns raised.

We look forward to receiving your revised manuscript.

Kind regards,

Ilse Bloom

Staff Editor

Journal Requirements:

Additional Editor Comments (if provided):

Reviewers' comments:

Reviewer's Responses to Questions

**Comments to the Author**

1. Does this manuscript meet PLOS Global Public Health’s publication criteria? Is the manuscript technically sound, and do the data support the conclusions? The manuscript must describe methodologically and ethically rigorous research with conclusions that are appropriately drawn based on the data presented.

Reviewer #1: Partly

2. Has the statistical analysis been performed appropriately and rigorously?

Reviewer #1: Yes

3. Have the authors made all data underlying the findings in their manuscript fully available (please refer to the Data Availability Statement at the start of the manuscript PDF file)?

Reviewer #1: Yes

4. Is the manuscript presented in an intelligible fashion and written in standard English?

Reviewer #1: Yes

5. Review Comments to the Author

Reviewer #1: The manuscript is well written and investigates the critical challenges in accessing timely radiological reports in a low-resource setting, using Ghana as a case study. The core claim is that systemic barriers, primarily a severe radiologist shortage, lead to prolonged report turnaround times. This, in turn, forces frontline clinicians to rely on their own, often limited, image interpretation skills, resulting in a cascade of negative outcomes: compromised diagnostic accuracy, delayed treatment, increased diagnostic uncertainty, and an elevated risk of misdiagnosis. The manuscript positions this issue not just as a logistical bottleneck, but as a critical failure point in the clinical care pathway that disproportionately affects low- and middle-income countries (LMICs).

However, its impact is limited by a lack of empirical data to quantify the scale and severity of the problem it describes.

The summary describes the findings but does not present the underlying data. A key weakness (e.g., actual average turnaround times, the frequency of report-less X-rays, rates of clinician-reported diagnostic uncertainty) to substantiate the powerful qualitative claims. All assertions about frequency and impact should be directly backed by the study's own data.

Without knowing the population size at each of the participating hospitals, there is a risk that the findings, while likely representative, may be overgeneralized for the entire country or region. All assertions about frequency and impact should be directly backed by the study's own data.

Key Areas Requiring Attention:

The manuscript describes findings but lacks the underlying data necessary to quantify the problem's scale. To substantiate its powerful claims, it is preferable if it could include more detail:

Quantitative metrics (if possible) such as average report turnaround times for different case urgencies.

Data on the frequency and percentage of X-rays provided without an accompanying report (if possible).

Systematically collected data on clinical impact, such as rates of diagnostic uncertainty or changes in treatment plan upon receiving the formal report (if possible).

Justification for the sample size and the selection of participating hospitals.

To ensure claims are appropriately scoped and supported:

All assertions about the frequency and impact of the described challenges (e.g., "clinicians regularly relied," "prolonged turnaround times remain a global challenge") must be directly backed by the study's own data.

The population sizes and clinical contexts of the participating hospitals should be provided to contextualize the findings and prevent overgeneralization for the entire country or region without sufficient evidence.

6. PLOS authors have the option to publish the peer review history of their article (what does this mean?). If published, this will include your full peer review and any attached files.

**Do you want your identity to be public for this peer review?** For information about this choice, including consent withdrawal, please see our Privacy Policy.

Reviewer #1: No

Figure Resubmissions:

---

## [Decision Letter · Decision Letter 1]

15 Mar 2026

PGPH-D-25-02975R1

Radiologist workforce challenges and the burden of image interpretation in Ghana: Implications for healthcare delivery and patient care

Dear Dr. Acheampong,

Thank you for submitting your manuscript to PLOS Global Public Health. After careful consideration, we feel that it has merit but does not fully meet PLOS Global Public Health’s publication criteria as it currently stands. Therefore, we invite you to submit a revised version of the manuscript that addresses the points raised during the review process.

We look forward to receiving your revised manuscript.

Kind regards,

Helen Howard

Staff Editor

Journal Requirements:

Additional Editor Comments (if provided):

Reviewers' comments:

Reviewer's Responses to Questions

**Comments to the Author**

1. If the authors have adequately addressed your comments raised in a previous round of review and you feel that this manuscript is now acceptable for publication, you may indicate that here to bypass the “Comments to the Author” section, enter your conflict of interest statement in the “Confidential to Editor” section, and submit your "Accept" recommendation.

Reviewer #1: All comments have been addressed

Reviewer #2: (No Response)

Reviewer #3: (No Response)

2. Does this manuscript meet PLOS Global Public Health’s publication criteria? Is the manuscript technically sound, and do the data support the conclusions? The manuscript must describe methodologically and ethically rigorous research with conclusions that are appropriately drawn based on the data presented.

Reviewer #1: Yes

Reviewer #2: Yes

Reviewer #3: Yes

3. Has the statistical analysis been performed appropriately and rigorously?

Reviewer #1: Yes

Reviewer #2: Yes

Reviewer #3: Yes

4. Have the authors made all data underlying the findings in their manuscript fully available (please refer to the Data Availability Statement at the start of the manuscript PDF file)?

Reviewer #1: Yes

Reviewer #2: Yes

Reviewer #3: Yes

5. Is the manuscript presented in an intelligible fashion and written in standard English?

Reviewer #1: Yes

Reviewer #2: Yes

Reviewer #3: Yes

6. Review Comments to the Author

Reviewer #1: Thank you for addressing all the comments and additional clarification.

Reviewer #2: Peer review report

Manuscript #: PGPH-D-25-02975R1

Thank you for the opportunity to review this interesting research. The manuscript has the potential to make a valuable and engaging contribution. However, several aspects of the work require further clarification and development before it is suitable for publication.

Title: The title effectively reflects the topic and its implications; however, it does not identify the study population. Clarify whether the perspectives or experiences are those of radiologists, radiographers, or clinicians, and incorporate this information into the title.

Abstract

• State whether participants are from the public, private, or both sectors, as this could influence outcomes.

• Please ensure consistency in terminology. In the aim, the term frontline clinicians (line 39) is used, whereas in the conclusion, front-line medical practitioners appear (Line 53). This inconsistency may confuse readers.

• Under the results line 224, some participants suggested the need to introduce reporting radiographers like other Western countries and in Africa, Uganda and Zimbabwe (Approved in November 2025). This is an important point which is missing in the abstract.

• Keywords should be arranged in alphabetical order (Line 57).

Introduction

• Line 82: The term “radiographers” is used; however, some are reporting radiographers trained in clinical reporting. Please be specific.

• The aim stated in the abstract does not align with the aim in the introduction. The abstract uses the verb “explore”, whereas the introduction uses “examine”. The research aim should be stated consistently throughout the paper. The research aim should be more explicitly stated at the end of the introduction.

• Please clarify whether image interpretation by referring clinicians in Ghana is limited to plain radiography (X-rays) or also includes advanced imaging modalities. This distinction should be clearly stated. This can be made in the introduction or discussion sections of the manuscript.

• Important contextual information appears to be missing regarding the education and training in radiology of referring clinicians. Including details on medical doctors’ training and experience in radiology and image interpretation, both theoretical and practical, would strengthen the argument. This may include time allocated to radiology training. Additionally, the shortage of radiologists may result in greater emphasis on clinical service provision (reporting) than on clinical teaching, which is relevant to the study context. This may affect future clinicians.

Materials and methods

• What is qualitative descriptive study? Why was it used?

State whether participants are from the public, private, or both sectors, as this could influence outcomes.

• The description of the recruitment process lacks sufficient detail, particularly regarding sampling strategies. Please clarify how participants were selected.

• It is unclear on what basis the interview guide was developed.

• The claim of data saturation is not justified. Provide more details. E.g., contacting additional interviewees to avoid early termination of data collection.

• The explanation of data analysis is incomplete. Please briefly outline all five relevant steps of Braun and Clarke’s six-phase thematic analysis framework. Additionally, clarify whether transcription and coding were conducted manually or using electronic tools.

• Data collection and data analysis should be presented as distinct sections. Consider writing about each in separate paragraphs for clarity. This also applied to ethical considerations. How did the researchers applied the 4 ethical principles of autonomy, beneficence, non-maleficence and justice?

• Trustworthiness strategies are not sufficiently described beyond peer debriefing. Include additional methods used to enhance credibility, dependability, confirmability, and transferability. You may use a table.

Results

• Line 134 does not address demographics but rather emerging themes. This content should be presented in a separate paragraph for clarity and logical flow.

• The themes are well presented; however, the supporting explanations are limited. Expanding the narrative would help readers better understand the findings and follow the progression of ideas.

• Consider presenting sub-theme names explicitly and consistently in the results section. For example: Sub-theme 1: Frequency of Imaging Requests. This approach would improve structure and readability.

Discussion

• Begin the discussion by restating the aim of the study to orient readers and frame the interpretation of the findings.

• Lines 289 to 300- This section focuses on the establishment of reporting radiographers in Ghana. Consider strengthening the discussion by providing examples from other settings where this practice has been implemented and outlining its benefits. For instance, in November 2025, the scope of practice for radiographers in Zimbabwe was expanded to include clinical reporting, making it the second country in Africa, after Uganda, to implement this model. Additionally, countries such as South Africa, Zambia and Nigeria are currently exploring the establishment of reporting radiographers. Including such comparisons would provide regional context and support the relevance of your findings.

• Education and training in radiology within medical schools require further discussion, as this appears to be a central issue. Consider comparing your findings with evidence from studies conducted outside Ghana to highlight similarities or differences in training experiences, competency development, and clinical preparedness.

Conclusion

• Among the strategies is the establishment of reporting radiographers. This is important to the development of quality imaging services.

Reviewer #3: thank you for this important work.

Abstract and introduction are clear and lead well into the study itself. There is a rationale for this study articulated.

The methodology is appropriate and the rationale for the sampling strategy is well addressed. Interviewer bias and means to ensure credibility are addressed. Researcher triangulation seems to also have been used, although not named explicitly as such. A good point is made regarding the sufficiency of the sample size and this does not need to be changed for this article, however, consideration might be given in future to the sub divisions of the sample and considering the sufficiency of the sample within these. Further detail on the researchers’ approach would be useful – how many authors generated initial codes? What was the procedure in cases of discrepancy? Was there an impartial researcher also who was not involved in the interviews? Were the interviews conducted with one researcher present? Were any field notes taken to support transcription? It would be useful to have the script available as part of the paper also.

Were there differences found in the perspectives of the various levels of experience of the doctors? It would be very useful to have a table to indicate the code and the year of experience of the clinician, however as there is only one interviewee in one experience band, I understand that this might not be appropriate.

Comments from the previous review have been addressed and there has been significant changes made to the original submission.

7. PLOS authors have the option to publish the peer review history of their article (what does this mean?). If published, this will include your full peer review and any attached files.

**Do you want your identity to be public for this peer review?** For information about this choice, including consent withdrawal, please see our Privacy Policy.

Reviewer #1: No

Reviewer #2: No

Reviewer #3: No

Figure Resubmissions:

---

## [Decision Letter · Decision Letter 2]

27 Apr 2026

Radiologist workforce challenges and the burden of image interpretation in Ghana: Perspectives of frontline doctors and implications for healthcare delivery

PGPH-D-25-02975R2

Dear MR Acheampong,

We are pleased to inform you that your manuscript 'Radiologist workforce challenges and the burden of image interpretation in Ghana: Perspectives of frontline doctors and implications for healthcare delivery' has been provisionally accepted for publication in PLOS Global Public Health.

Best regards,

Julia Robinson

Executive Editor

Reviewer Comments (if any, and for reference):

Reviewer's Responses to Questions

**Comments to the Author**

1. If the authors have adequately addressed your comments raised in a previous round of review and you feel that this manuscript is now acceptable for publication, you may indicate that here to bypass the “Comments to the Author” section, enter your conflict of interest statement in the “Confidential to Editor” section, and submit your "Accept" recommendation.

Reviewer #2: All comments have been addressed

Reviewer #3: All comments have been addressed

2. Does this manuscript meet PLOS Global Public Health’s publication criteria? Is the manuscript technically sound, and do the data support the conclusions? The manuscript must describe methodologically and ethically rigorous research with conclusions that are appropriately drawn based on the data presented.

Reviewer #2: (No Response)

Reviewer #3: Yes

3. Has the statistical analysis been performed appropriately and rigorously?

Reviewer #2: (No Response)

Reviewer #3: Yes

4. Have the authors made all data underlying the findings in their manuscript fully available (please refer to the Data Availability Statement at the start of the manuscript PDF file)?

Reviewer #2: (No Response)

Reviewer #3: No

5. Is the manuscript presented in an intelligible fashion and written in standard English?

Reviewer #2: (No Response)

Reviewer #3: Yes

6. Review Comments to the Author

Reviewer #2: I have carefully reviewed the revised version of the manuscript. The authors have addressed all previously raised concerns and implemented the necessary corrections thoroughly. In my opinion, the manuscript is now suitable for publication, and I recommend its acceptance.

Reviewer #3: Thank you for this revised version of the manuscript. Chages have been appropriately made. I feel that the interview script (that is, the questions asked to the interviewees, rather than the transcript of their responses), should be included. I feel that this will strengthen the work. All other comments of mine are addressed. Thank you.

7. PLOS authors have the option to publish the peer review history of their article (what does this mean?). If published, this will include your full peer review and any attached files.

**Do you want your identity to be public for this peer review?** For information about this choice, including consent withdrawal, please see our Privacy Policy.

Reviewer #2: No

Reviewer #3: **Yes:** Dr Clare Rainey
